# Views and experiences of maternal healthcare providers regarding influenza vaccine during pregnancy globally: A systematic review and qualitative evidence synthesis

**Fatemah Alhendyani**[1,2]*, **Kate Jolly**[1], **Laura L. Jones**[1]

1 Institute of Applied Health Research, University of Birmingham, Birmingham, United Kingdom,
2 Department of Public Health, Ministry of Health, Kuwait, State of Kuwait

* f.alhendyani@gmail.com

## Abstract

### Background

The World Health Organization (WHO) recommends that pregnant women receive influenza vaccination; however, uptake of the vaccine remains low. Maternity health care professionals (MHCPs) play an important role in motivating pregnant women to receive the influenza vaccine. However, factors such as MHCPs' views and knowledge about the vaccine, and time constraints due to workload may influence MHCPs' practices and opinions about women receiving the influenza vaccine during pregnancy. To date, the qualitative evidence exploring MHCPs' views and experiences around influenza vaccine uptake in pregnant women has not been synthesised.

### Aim

To systematically review and thematically synthesise qualitative evidence that explores the views and experiences of MHCPs involved in the provision of the maternal influenza vaccine worldwide.

### Methods

Five databases (MEDLINE, EMBASE, PsycINFO, CINAHL, Web of Science) were searched, supplemented with searches of included paper reference lists and grey literature. Study selection was conducted by up to three researchers applying pre-specified inclusion/exclusion criteria. Quality assessment was undertaken, data were extracted, coded and synthesised to develop descriptive and analytical themes.

### Results

Eight studies involving 277 participants were included. Seventeen descriptive themes were interpreted, embedded within six analytical themes. MHCPs perceived that maternal influenza vaccination delivery can be facilitated by trusting relationships, good communication, knowledge about the vaccine leading to confidence in recommending vaccine, electronic

**Data Availability Statement:** All relevant data are within the manuscript and its Supporting Information files.

**Funding:** The author(s) received no specific funding for this work. KJ is supported by the National Institute for Health Research (NIHR) Applied Research Centre (ARC) West Midlands. However, the views expressed are those of the author(s) and not necessarily those of the NHS, the NIHR or the Department of Health and Social Care.

**Competing interests:** The authors have declared that no competing interests exist.

vaccination prompts, and presence of national guidelines. However, workload, time constraints, MHCP's perception of pregnant women's concerns, and social/cultural/environmental influences could prevent the likelihood of delivery of influenza vaccine. Knowledgeable MHCPs who were regularly updated about vaccination based on scientific evidence were more confident when discussing and recommending the influenza vaccine to pregnant women. In addition, the presence of national policies and guidelines and electronic prompts for maternal influenza vaccination would enhance the delivery of the vaccine.

## Conclusion

Our findings suggest that approaches to enhance the vaccination uptake rate in pregnant women include addressing MHCPs barriers to discussing influenza vaccination through education, sufficient time for discussions, and electronic prompts about vaccination, as well as evidence based local and national guidelines.

## 1 Introduction

Pregnant women have greater risk from influenza infection than other population groups [1]. They are more likely to be hospitalised due to influenza than non-pregnant women [2–7], especially those with chronic medical conditions [8,9]. Moreover, infants born to women who develop influenza during pregnancy have a higher risk of low five-minute APGAR (appearance, pulse, grimace, activity and respiration) scores [10]. They also have a higher rate of low birth weight [4,5,10]. A recent study showed that influenza infection during pregnancy was not associated with an increased risk of preterm birth (adjusted Hazard Ration (aHR) 1·4, 95% confidence interval (CI) 0·9 to 2·0; p = 0·096) but was associated with an increased risk of late pregnancy loss (aHR 10·7, 95% CI 4·3 to 27·0; p<0·0001) [11].

Since 2012, the World Health Organization (WHO) has recommended annual influenza vaccination for pregnant women [1]. However, not all countries have a national program for influenza immunization. In 2014, only 59% of the 194 WHO Member States had a national influenza immunization policy for general population [12]. The inactivated influenza vaccine (IIV) has been reported to be effective in reducing laboratory-confirmed influenza rates for pregnant women [8–10]. Dabrera et al. [13] reported that the IIV was 71% effective (95% CI: 24–89%) at preventing influenza infection and 64% effective (95% CI: 6–86%) at preventing influenza hospitalisation for pregnant women. Also, evidence showed the vaccine offered immunity to the infant for up to six months after birth and preventing about 30–63% of laboratory-confirmed influenza cases in infants less than six months of age [9].

Despite the WHO recommendations, refusing vaccination during pregnancy is common globally, resulting in a low vaccine coverage rate [14–18]. There are two recommended vaccines during pregnancy including influenza vaccine and Tdap (Tetanus, diphtheria, and pertussis) vaccine. In 2019–20 the United States (US) reported that 61.2% of pregnant women received influenza vaccine, 56.6% received Tdap, and 40.3% received both vaccines [19]. A 2020 expert commentary review reported heterogeneity of coverage globally from less than 10% coverage in Europe, to 50–60% in US, and >95% in Brazil [20].

Maternity healthcare professionals (MHCPs) play an important role in offering the influenza vaccine and influencing the decisions of pregnant women in relation to uptake [21–24]. However, MHCPs knowledge of and beliefs about the influenza vaccine, and their experiences

supporting pregnant women, as well as their workload and time constraints, affect their practices in advocating for influenza vaccine during pregnancy [21–24].

Most studies of the factors influencing the uptake of the maternal influenza vaccine have focused on the perspectives of pregnant women, addressing issues such as their knowledge, beliefs, attitudes and behaviours towards influenza vaccination during pregnancy rather than the perspectives of MHCPs [15,18,25–27]. Most of the studies that have addressed the perspectives of MHCPs toward maternal influenza vaccine have used cross-sectional survey designs [28–32] and have not explored their views and experiences qualitatively.

Questions such as how often pregnant women are being offered the influenza vaccine or how often pregnant women are receiving the influenza vaccine need to be addressed. This review largely focusses on the offer of the vaccine. The aim of the study was to systematically review and thematically synthesise qualitative evidence that explores the views and the experiences of MHCPs involved in the provision of the maternal influenza vaccine worldwide. In addition, we identified and analysed interventions that MHCPs consider feasible to implement to increase vaccination uptake.

## 2 Methods

Reporting of this review was guided by the Preferred Reporting Items for Systematic Reviews and Meta-Analyses (PRISMA) checklist [33] provided in S1 Table and Enhancing Transparency of Reporting the Synthesis of Qualitative Research (ENTREQ) framework [34] provided in S2 Table. This review was registered with the International Prospective Register of Systematic Reviews, PROSPERO, registration number CRD42020187564.

### 2.1 Search strategy

The sample, phenomenon of interest, design, evaluation and research type (SPIDER) tool [35] was used to create a clearly defined review question and to support the development of the search strategy. Search terms were created based on concepts identified in the question formulation, including vaccination, antenatal care, pregnancy, healthcare professional, maternity, vaccine hesitancy and practice. Comprehensive search of five bibliographic databases were conducted to include articles published between January 2012 and March 2020: MEDLINE; EMBASE; CINAHL; PsychINFO and Web of Science. Both index headings, such as Medical Subject Heading (MeSH) terms, and free text words were searched to ensure specificity and sensitivity of the search strategies [36]. Additional hand-searches were conducted based on included studies' reference lists and citations (in Google Scholar) as well as grey literature, such as OpenGrey, were searched. However, due to resource limitations, non-English studies were excluded from this review. An example search strategy is given in supplementary S3 Table.

### 2.2 Study selection

All search results were imported into Endnote v.X9 (Clarivate Analytics, Philadelphia, US). Duplicates were removed electronically and manually. Following this titles, abstracts and full text papers were independently screened based on a pre-specified inclusion/exclusion criterion (Table 1) by two of three authors (FA and KJ/LLJ). Discrepancies in the screening process were resolved via discussion and/or a third reviewer.

### 2.3 Data extraction

Study characteristics were extracted by one researcher (FA) based on a pre-developed data extraction proforma. Information extracted included the study origin (e.g., country),

**Table 1. Inclusion/Exclusion criteria.**

| | Inclusion criteria | Exclusion criteria |
|---|---|---|
| **S**ample | • Maternity healthcare professionals (e.g., general practitioners dealing with pregnant women, nurses, obstetricians, gynaecologists, pharmacists, midwives, health workers and public health personnel working in maternity healthcare settings e.g. community health workers working in antenatal settings) | • Pregnant women including healthcare professionals who are pregnant |
| **P**henomenon of **I**nterest | • Influenza vaccine during pregnancy | • Other maternal vaccines (e.g., Tdap) |
| **D**esign | • Qualitative data collection methods (e.g., interviews/focus groups)<br>• Mixed methods where qualitative element can be isolated for synthesis<br>• Qualitative questionnaire data | • Quantitative study designs (e.g. cross sectional surveys) |
| **E**valuation | Views, attitudes, beliefs, knowledge, opinions, practice, perceptions and experiences | |
| **R**esearch type | • Qualitative<br>• Mixed methods | • Conference abstracts<br>• Editorials<br>• Quantitative studies<br>• Intervention studies<br>• Systematic reviews<br>• Auto-ethnographic papers |
| Limitations | • English language | • Full text could not be obtained through institutional access<br>• Qualitative primary research based on pregnant women's and maternity healthcare professionals' context where the latter could not be disaggregated for synthesis<br>• Qualitative primary studies had a mix of different participants (e.g. commissioners and MHCPs) where the data for MHCPs could not be disaggregated for synthesis. |

MHCP: Maternity healthcare professional.

Tdap: Tetanus, diphtheria, pertussis.

healthcare setting (e.g., primary care or tertiary care), sampling (e.g., purposive or snowballing), participants' information/characteristics (e.g., doctor or midwife), sample size, data collection method (e.g., interview or focus group), analysis method (e.g., thematic analysis, content analysis or framework analysis), study aim, findings and recommendations. The findings (results) and discussion sections of the included articles were imported into NVivo v.12 software (QSR International Pty Ltd, Melbourne, Australia) for analysis.

## 2.4 Quality assessment

The included studies were assessed by two researchers (FH and LLJ) following a modified Critical Appraisal Skills Programme (CASP) checklist for qualitative research [37]. Studies were assessed for their clarity, appropriateness, rigour of methodological reporting, ethical considerations and reflexivity. Currently, there is no agreed consensus method for quality assessment in the synthesis of qualitative research [38,39]; all studies were included irrespective of their reporting quality given that they contributed to the conceptual richness of the synthesis [39].

## 2.5 Data analysis and synthesis

A thematic synthesis was conducted following the methods outlined by Thomas and Harden [40]. A data-rich article was selected as the index article [24]; findings (results) and discussion sections were coded as well as quotations, author's commentary and interpretations. Other articles were then coded by applying the same method in descending order of data richness. Concepts in each article were coded to iteratively develop a codebook, with each article having

an ability to contribute new codes. The codes were examined and discussed multiple times between the authors to refine the developing codebook. The codebook was analysed to inform descriptive themes closely resembling the prevailing concepts across primary studies. A conceptual model was then developed to build higher order concepts within which analytical and descriptive themes were located. In the results, we have distinguished the primary quotes in italic from author interpretation which is typed in normal font.

## 3 Results

### 3.1 Systematic search and selection

Systematic database searches identified 6316 articles. After removal of duplicates, 308 studies remained. A total of 295 articles were excluded based on inclusion/exclusion criteria applied to titles and abstracts. Full texts of the remaining 13 studies were sought for detailed evaluation against the inclusion criteria. After reviewing the full-texts and applying the selection criteria eight articles were included in the thematic synthesis. A flow diagram of the search strategy is illustrated in Fig 1.

### 3.2 Characteristics of included studies

The studies represented different contexts and locations, including high-, middle- and low-income countries. Three articles were from Australia [24,41,42], and one from the USA [43], Kenya [44], El Salvador [45], the UK [46] and China [47]. Articles were published between 2014 and 2019. There were seven qualitative studies [24,41–44,46,47] and one mixed methods study [45]. All qualitative studies used thematic analysis, four combined a hybrid deductive/inductive approach [41,43,44,46], while four used an inductive approach using constant comparative methods [24,42,45,47].

Studies included data from 277 participants including nurses, midwives, general practitioners (GPs), obstetricians, practice nurse, practice managers, community leaders, community health personnel, public health managers and experts (national and international levels), policymakers, clinical officers and physicians. Purposive sampling was used by seven studies [24,42–47] and snowballing sampling by one [41]. Data collection methods were interviews, three in-depth [43,46,47] and five semi-structured interviews [24,41,42,44,45]. Table 2 summarises characteristics of included articles and full details of data extraction are provided in S4 Table.

### 3.3 Quality assessment

The critical appraisal of the included studies highlighted that the reporting quality was generally good. All eight studies provided a clear statement of aims and used appropriate qualitative methods. The research design was explicitly justified in all but two studies [44,47]. Recruitment strategy was discussed in all but one study [44]. The relationship between researcher and participants was not adequately considered in five studies [42,44–47]. Ethical considerations had been reported in five studies [24,41,44,45,47]. All studies had conducted a rigorous analysis. Findings were clearly discussed in the context of wider research literature, policy and practice among all studies. Transferability was discussed in all studies but one [46]. Full details of the CASP assessment are provided in S5 Table.

### 3.4 Thematic synthesis findings

Views and experiences of MHCPs about influenza vaccine during pregnancy were interpreted within 17 descriptive themes, embedded in six analytical themes. Table 3 provides core

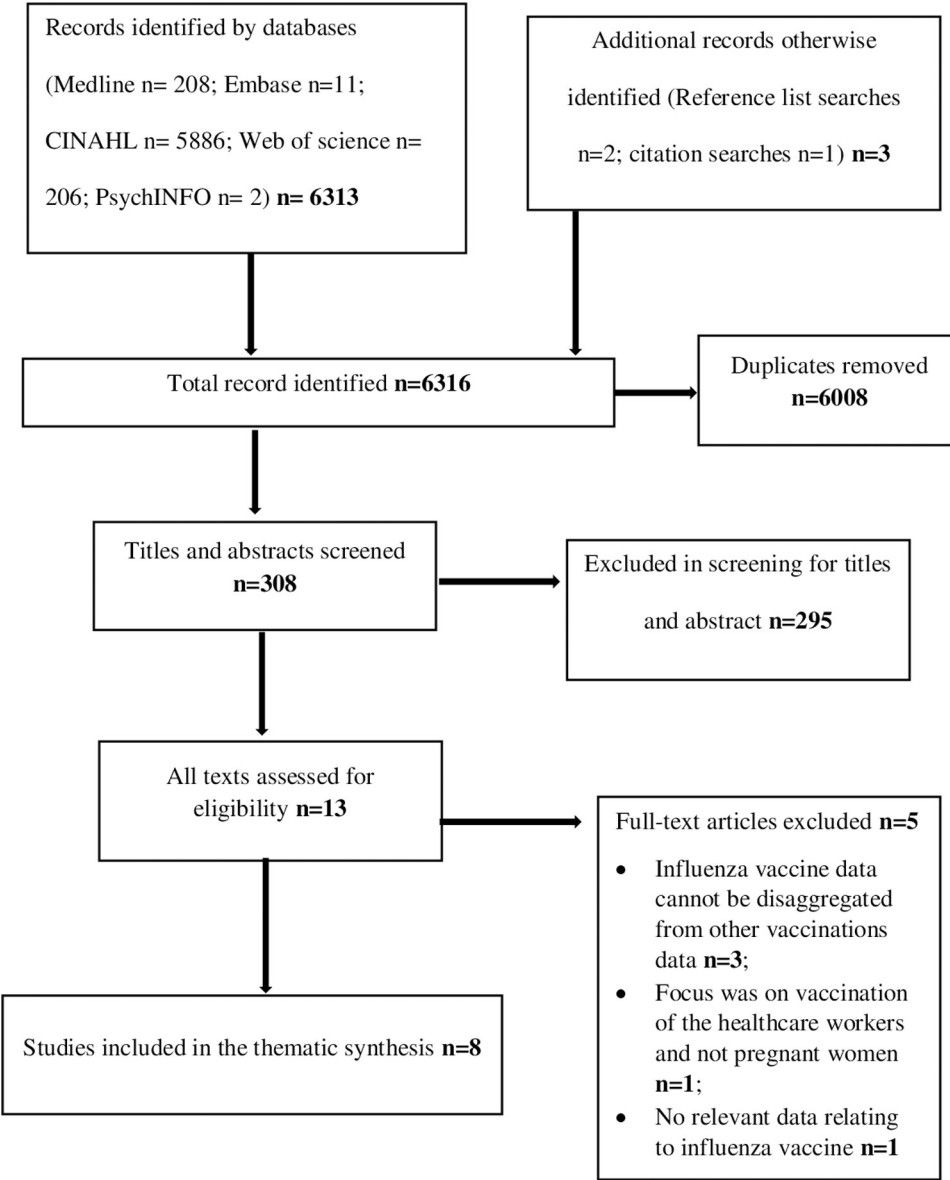

**Fig 1. Flow diagram of systematic search and study selection.**

analytical themes and descriptive themes highlighting MHCPs' experiences and views relating to the provision of influenza vaccine for pregnant women, this table is further explained, with evidence, in the accompanying writing. All quotes included below are from HCP participants from the respective articles.

## 3.5 Conceptual model

The themes are drawn together in a conceptual model, illustrated in Fig 2.

**3.5.1. Social, cultural and environmental influences.** Social and cultural norms, including religion, family involvement, media influence and access issues, were perceived by MHCPs to influence a pregnant woman's decision about influenza vaccination in pregnancy or access to antenatal care and thus the opportunity to be offered influenza vaccination.

**Table 2. Characteristics of studies included in the thematic synthesis.**

| Author | Country and publication year | Healthcare setting | Participants and sample size | Study aim | Methodology | Analysis method |
|---|---|---|---|---|---|---|
| **Maher et al [24]** | Australia, 2014 | GP practices | 17 general practitioners | Explore knowledge, attitude, beliefs and practices of GPs about maternal influenza vaccine. | Semi-structured interviews | Thematic analysis |
| **Kaufman et al [41]** | Australia, 2019 | Tertiary hospitals | 12 midwives | Explore midwives' attitudes, values and delivery of maternal vaccine. | Semi-structure interviews | Thematic analysis, hybrid approach |
| **Webb et al [42]** | Australia, 2014 | Tertiary hospitals | 15 GPs, obstetricians, and midwives | Explore the practice of HCPs' maternal vaccine uptake, knowledge, attitudes, beliefs, and practice. | Semi-structure interviews | Thematic analysis |
| **Frew et al [43]** | USA, 2018 | Obstetrics-genecology practices | 24 physicians, nurse practitioners, midwives, medical assistants and practice managers | Identify attributes and facilitators for vaccine intervention in obstetrics-gynecology settings. | In-depth interviews | Thematic analysis using hybrid approach |
| **Bergenfeld et al [44]** | Kenya, 2018 | Not specified, public facilities | 111 nurses and clinical officers | Investigate acceptance and demand for maternal vaccines in Kenya from HCPs perspective. | Semi-structured interviews | Thematic analysis using hybrid (deductive and inductive) approach |
| **Fleming et al [45]** | El-Salvador, 2018 | Not specified | 70 community (leaders, health personnel), public health managers and experts, policymakers, physicians | Share experiences from El-Salvador about maternal influenza vaccine, as it has high coverage rate. | Key informant interviews and semi-structured interviews | Thematic analysis |
| **Wilson et al [46]** | UK, 2019 | GP practices | 10 GPs, midwives and practice nurses | Understand access to, and attitudes towards maternal vaccination among HCPs. | In-depth interviews | Thematic analysis-hybrid approach |
| **Li et al [47]** | China, 2018 | Tertiary hospitals | 18 obstetricians | Understand HCPs' perception and attitude toward maternal influenza vaccine. | In-depth interviews | Thematic analysis, inductive, constant comparative method |

In some settings, MHCPs perceived certain religious denominations restricted pregnant women from accessing antenatal care visits and hence uptake of the influenza vaccine [44]. In Kenya, when MHCPs were asked about what intervention they thought would help to address the religious restrictions toward vaccination, MHCPs proposed attending Sunday church services to discuss health concerns and provide information to pregnant women there to reach a larger demographic [44]:

> *"I asked her why she didn't want to go for ANC services for her first born! I urged her that it was important to get vaccinated. She told me that their church doctrine restrains them from going to hospital. They get such instructions from their pastors."* [Kenya,44]

They reported that in societies where men have dominance over women, pregnant women were often restricted from receiving antenatal care by their husbands, which may have resulted in low rates of influenza vaccination's uptake [44]. In societies where MHCPs found it difficult to persuade pregnant women to receive medication including vaccine, other family members who have power over these women's decisions, such as mothers-in-law, were prohibiting pregnant women to take the vaccine [47]:

> *"She got married to guy who prevented her from going for [antenatal] clinic."* [Kenya,44]

**Table 3. Core analytical and descriptive themes for MHCPs' experiences and views relating to the provision of influenza vaccination for pregnant women.**

| Analytical themes | Descriptive themes | Codes |
|---|---|---|
| Social, cultural and environmental influences | Religion | • Religious beliefs against vaccination [44]. |
| | Family influence | • Male dominance [44]. <br> • Other family members attitude toward vaccination [47]. |
| | Media influence | • Anti-vaccination campaigns [43]. |
| | Access issues | • Areas of high criminal rates [45]. <br> • Long distance travel required to healthcare facilities [44]. |
| MHCPs perceived views of pregnant women's health literacy and beliefs | Hesitancy | • Concerns about safety and effectiveness of the vaccine [42,47]. |
| | Health literacy | • Lack of information about influenza infection as well as the influenza vaccination [41,43,44]. |
| Interaction with MHCP | Trust and social position | • Social position of MHCP [44]. <br> • Good relationship with pregnant woman [24,44]. |
| | MHCP communication and provision of education to pregnant women | • Based on pregnant women beliefs [24,43,46]. <br> • Communication approach [41,43]. <br> • Workload and time-limit [42,44]. <br> • MHCP responsibility to educate pregnant women [42,43]. |
| | Pregnant women's informed choice | • Support woman's choice [24,43,46]. |
| MHCPs knowledge and attitude | Knowledge | • Influenza infection [24,41,44]. <br> • Influenza vaccination [24]. <br> • Recommendations for pregnant women vaccination [24,41]. |
| | Attitude | • Concerns about safety [24,45,47]. <br> • Not convinced about prioritising pregnant women [45,47]. |
| Local healthcare system | Training and education for MHCP | • Recent recommendations [44]. <br> • Training on vaccination administration [24,41,44]. <br> • Based on scientific evidence [24,44]. |
| | Dedicated vaccination team | • Informed about recommendations [41–43,46]. <br> • Responsible for update in vaccination guidance [24,42]. <br> • Deliver vaccine to pregnant women [42,43]. |
| | Record keeping and vaccination prompt | • History of vaccinations [42]. <br> • Prompt: sticker, checklist, electronic [24,41,43]. |
| National policy and practice | Policy and guidelines | • Part of national policy [24,42,47]. |
| | Centralisation of information system | • Sharing information system [44]. |
| | Public vs. private provision | • Profiting from vaccination [44,45]. |

MHCP: Maternity healthcare professional.

*"Sometimes, you just convince the pregnant women with a lot of effort, and then the mother-in-law won't agree to let her take the medicine"* [China,47]

In addition, anti-vaccine campaigns in the media can negatively influence the likelihood that a pregnant woman will receive vaccines [43]:

*"Heard through the media, negative things through the media. So, what I have heard is 'Oh, well. . .'"* [USA,43]

Geographical barriers to accessing healthcare services, and hence access to antenatal care including vaccination, including one example of a high crime rate area deterring pregnant women from attending clinics or necessitating police support to do so, as well as long distances and a lack of transportation to clinics. For instance, in rural areas, pregnant women avoided going to a health facility because of the long distance, which may have resulted in low uptake of influenza vaccines in those areas [44,45]:

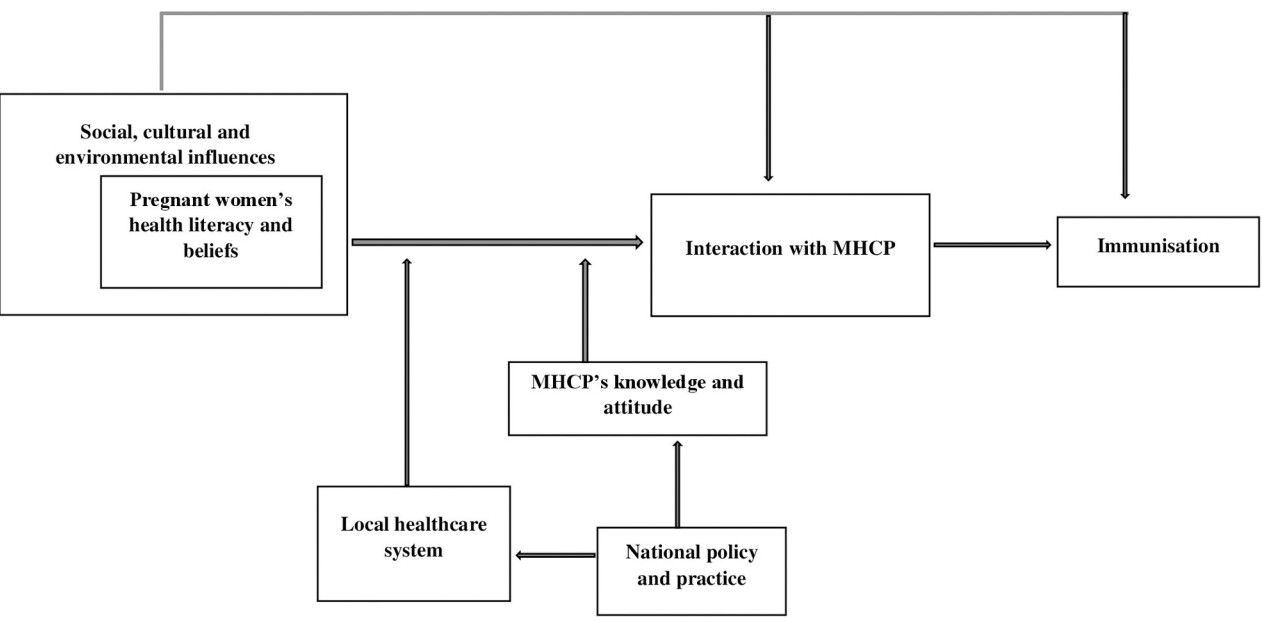

**Fig 2. Illustrative model of analytical and descriptive themes.** MHCP: Maternity healthcare professional.

*"...There are a lot of municipalities with crime problems."* [El-Salvador,45]

*"Distance to health facilities and lack of transportation was one of the major access issues perceived by providers to hinder maternal vaccine coverage"* [Kenya,44]

**3.5.2. MHCPs perceived views of pregnant women's health materials and beliefs.** Across the studies, MHCPs discussed the factors that may influence pregnant women's decisions toward the uptake of the influenza vaccine including hesitancy and lack of health literacy.

Despite the availability of influenza vaccine, MHCPs reported that pregnant women have more concerns about the influenza vaccine than the pertussis vaccine and are hesitant to receive the former due to a lack of awareness about the recommendations as well as concerns about the safety of the vaccine [41]. In addition, the idea of needing a vaccine to prevent influenza infection was not accepted by pregnant women who reported that the infection can be managed by natural remedies [47]:

*"Women are maybe more hesitant to get the flu compared to pertussis...They think they're going to have more reactions to it"* [Australia,41]

*"They stated that it would be even more difficult to convince pregnant women to accept a vaccine merely to prevent flu"* [China,47]

MHCPs stressed the need for health materials about influenza vaccine during pregnancy in the form of posters, brochures or education hubs placed in health care facilities or provided to pregnant women to eliminate their concerns and promote vaccination [41,43,44]. Additionally, they discussed the need for health literacy that is translated into different languages and broadcasted through community radio stations [44]:

*"I think if there was, like, an education hub type thing for vaccinations in pregnancy"* [Australia,41]

*"I think the information should be printed out in different languages and given to the community radio stations so that it can be explained to them."* [Kenya,44]

**3.5.3. Interaction with MHCP.** Trust between MHCPs and pregnant women as well as MHCPs' social position, communication approach to educating pregnant women, and pregnant women's informed choices were interpreted as themes across the papers.

MHCPs reported that their ability to convince pregnant women to receive influenza vaccine was improved by having a good and trusting relationship with the pregnant woman [24,44].

*"I have a good relationship with the patients and if I recommend it they probably would take it on board"* [Australia, 24]

Bergenfeld et al. [44] explained that because of the MHCP's authority and superior knowledge, pregnant women were more likely to trust MHCPs and follow their recommendations:

"*Maybe sometimes they do not have that chance to say no because they look at me as their savior. . . and everything I tell them, they believe is right*" [Kenya,44]

MHCPs should be able to effectively communicate and provide information/education to pregnant women [42,43]. MHCPs approached the introduction and promotion of influenza vaccination through one-to-one discussions to address pregnant women's concerns and allowing them to ask questions [43]:

*I will definitely base the conversation on kind of how they're phrasing their questions and what their questions are. So I try to answer the question, and then may ask a couple questions back to just get a feel for what they already know about vaccinations, or what they already believe about it. And then we'll kind of reframe it based on their system of beliefs.* [US,43]

Using passive language in the discussion with pregnant women to frame the significance of influenza vaccination during pregnancy, using this language with pregnant women makes them feel that taking the vaccine is the right thing to do as it is of advantages for themselves and their new-borns [41]:

*"It is recommended. The research shows that it is advantageous."* [Australia,41]

Additionally, influenza vaccination administration was framed as a choice for pregnant women rather than the MHCPs' recommendation. MHCPs' role was to provide advice only, rather than share in the decision making [24,43,46]. However, Wilson et al. [46] reported that influenza vaccination was not recommended by some MHCPs but merely mentioned:

*"We do not push it. We do not insist. We just advise them. If they accept that's fine"* [Australia,24]

Workload and time limit were factors reported by MHCPs as limiting influenza vaccination discussion with pregnant women [42,44]:

*"Maybe when a health care provider is in a hurry or is being overworked, you may find a long queue at the ANC waiting for vaccination. The nurse there may not have time to discuss much with every client about the vaccines. Sometimes they issue orders for the mothers to queue and get vaccinated. These are situations which may happen when there are several mothers at the clinic. This can cripple vaccine uptake since there is no time for explanations"* [Kenya,44]

*"The doctors are very busy and so for example in the private clinic we run at capacity so we're turning women away, so we basically have to apply um err almost triage principles to how we run our consultations but we do have a um midwife there with us who is our personal sort of assistant if you like. So things like breastfeeding and err analgesia in labor and vaccinations although I don't know if they mention vaccinations I'll be honest. We tend to delegate to them. The longer we make the consultations basically the less patients we can see."* [Australia,42]

**3.5.4. MHCP's Knowledge and attitude.**   MHCP knowledge and attitude about influenza vaccine during pregnancy were identified within this theme. MHCPs showed variations in their perceptions of the risks associated with influenza infection during pregnancy [24,41,44]. There was a lack of knowledge about influenza vaccination recommendations during pregnancy [24]. Thus, influenza vaccine did not appear to be a priority among MHCPs compared to other vaccines, and when it is promoted, it is to protect the mother rather than the foetus [41]:

*"I think with the number of people (pregnant women) who catch the flu and the number of people who don't have any problems with it. . ..I see it's a small amount of risk involved"* [Australia,24]

*"Generally, pertussis vaccination was presented in terms of protecting the baby, while influenza was primarily to protect the mother."* [Australia,41]

Three studies included discussions of MHCPs' attitudes [24,45,47]. Some MHCPs expressed that they need to assure themselves about a vaccine before recommending it to pregnant women [44]. Others reported that there are many strains of influenza viruses, so they had doubts about the effectiveness of the vaccine [47]:

*"We, the health care personnel, are the main factor [in low vaccination acceptance]. . .first, we have to be convinced ourselves in order to convince the population."* [El-Salvador,45]

**3.5.5. Local healthcare system.**   In order to enhance the uptake of influenza vaccine during pregnancy, MHCPs recommended professional training and education for them, dedicated vaccination teams, improved record keeping and electronic vaccination prompts.

Professional education and training workshops or campaigns for MHCPs would help to promote the discussion of maternal influenza vaccination by increasing MHCPs' knowledge and hence their confidence [41,44]. Continuous professional education about vaccinations other than tetanus toxoid was needed among MHCPs to provide details on vaccines' side effects and efficacy [24]. This approach would prepare MHCPs to be better advocates for vaccinations and to address pregnant women's concerns and hesitations:

*"If you do not update me properly, it will be of no use even if [a new vaccine] is brought to the facility. It will be available but I will not give it out, not because the patients are not asking for it, but because it is me who is not interested in giving it out"* [Kenya,44]

*"Those who had completed additional training reported feeling confident in their knowledge and ability to discuss vaccines"* [Australia,41]

A lack of standardisation for MHCPs regarding influenza vaccine delivery during pregnancy was reported in four studies [41–43,46]. Some MHCPs felt that they were not responsible for discussing influenza vaccination with pregnant women and that this is their GP's responsibility [42]. However, MHCPs reported that vaccination discussion and delivery is a nurse's responsibility, as they have more time to discuss this with pregnant women [43]:

*"We don't have any role in that [maternal vaccination]. We don't organize that, I usually send them off to their GP. If you want influenza vacs you can get it through your GPs the best place."* [Australia,42]

*"A nurse 'cause they would have some clinical knowledge and probably the time to do it. I don't see a physician doing that here. Just as far as timing. Um, and then a practice manager, I don't know that they would have all the clinical knowledge, and all the information."* [USA,43]

The need for a vaccination champion who is managing and creating enthusiasm for vaccination campaigns as well as encouraging better communication about vaccination between MHCPs, was discussed in two studies; it was perceived that vaccine champions may increase the delivery of maternal influenza vaccine [43,46].

In addition, MHCPs suggested using reminders such as a checklist, sticker or electronic prompt to discuss the influenza vaccine with pregnant women [24,41,43]. Also, they reported an absence of influenza vaccination's history reference in documentation and lack of mechanism for documenting the uptake of the vaccination or to follow up [Australia,42]:

*"In addition, immunization history is not part of the lengthy medical, psycho-social, surgical, and family history taken at a woman's first antenatal visit."* [Australia,42]

**3.5.6. National policy and practice.** Setting a national policy and developing guidelines, centralizing vaccine information and providing influenza vaccine for free would enhance MHCPs provision of influenza vaccination discussion, recommendation and delivery during pregnancy.

A lack of policies and guidance on vaccinations has resulted in some MHCPs being unaware about recommendations for influenza vaccine delivery during pregnancy [24,42]. Although, in countries where vaccination policies did exist, some MHCPs were not informed about existing policies and guidelines [47]:

*"Yes, well, it was recommended from the health department to do so I would assume that the information is accurate and there's no risk in doing it so I'm happy to follow that"* [Australia, 24]

*"We have never been informed or required to recommend influenza vaccine to pregnant women. We will do the work only when higher level administrative sectors require us to do so."* [China,47]

Pregnant women's geographical mobility and lack of a consistent information system made it difficult for MHCPs to keep track of pregnant women's vaccination history [44]:

*"This was linked to the notion that women often move during their pregnancy, leaving gaps in their medical records that are difficult to address without an integrated system."* [Kenya,44]

MHCPs reported that the private sector's profiting from influenza vaccinations may affect the delivery of the vaccine [44]. Therefore, MHCPs sent pregnant women to get their vaccines from free public hospitals [45]:

*"Private hospitals are established to make profits. The charges may keep them away"* [Kenya,44]

## 4. Discussion

### 4.1 Summary of findings and integration with literature

We believe this is the first systematic qualitative evidence synthesis of the views and experiences of MHCPs in relation to influenza vaccination during pregnancy. The main findings of this review indicate that the MHCPs' decision to recommend the influenza vaccine is framed by socio-cultural influences as well as MHCPs perceptions of pregnant women's beliefs. MHCPs' interactions with pregnant women play an important role in influencing their decisions regarding vaccination, and the MHCPs' knowledge about vaccination and attitudes towards it guide these interactions. A systems approach would address the lack of documentation of influenza vaccination and lack of a documentation or auditing system for vaccination intake and to follow up. The range of different MHCPs who could be involved in promoting and delivering influenza vaccination can lead to boundary issues as to whose responsibility it is. Direct policies and guidelines to ensure that MHCPs are regularly updated about vaccinations will lead to knowledgeable and confident MHCPs, thus increasing the likelihood of maternal influenza vaccine uptake. Moreover, MHCPs discussed strategies to increase pregnant women's knowledge and vaccine uptake, such as visiting religious services in Kenya, and creating educational hubs for pregnant women. Suggestions to increase delivery of the vaccine included vaccine champions, creating electronic vaccination prompts, system level informatics regarding who is due an influenza vaccination, free vaccine and a dedicated vaccine team. Our findings may have insights for the provision of vaccination against COVID-19 during pregnancy such as knowledge and beliefs perceived by the MHCPs toward the delivery of COVID-19 vaccination.

**Barriers to maternal influenza vaccination as perceived by MHCPs.** Provided that a woman was able to access antenatal care, then barriers to influenza vaccination included MHCPs' workload, time constraints, MHCP's perceptions of women's concerns about influenza vaccine and a lack of knowledge on the part of MHCPs in relation to influenza vaccine.

Social, cultural and environmental influences, including religion, have been widely discussed within the evidence base as a barrier to vaccination in general [48–51]. The religious objection against vaccination has many explanations, such as interference with the natural order by restricting the natural course of events, or because the vaccine itself does not meet religious dietary laws [51].

Anti-vaccination campaigns have been shown to undermine the authority of HCPs toward vaccinations [43]. A 2018 study of 314 pregnant women found that 22% showed hesitancy toward the influenza vaccine because of negative campaigns in the media [16].

MHCPs' knowledge varied across the different studies in our synthesis. For example, Maher et al. [24] showed that limited knowledge about influenza vaccination among MHCPs would result in either not recommending the vaccine or recommending it with varying levels of precision and confidence.

Workload and time constraints are two main barriers to influenza vaccination discussion and delivery among HCPs. The results of this qualitative evidence synthesis reflect evidence from previous quantitative studies; because influenza vaccination is not a priority for most HCPs, workload and time constraints have led to neglect of its discussion and recommendation [22,52].

**Facilitators to maternal influenza vaccination as perceived by MHCPs.**   Facilitators to influenza vaccination included trust and a good rapport between MHCP and pregnant women, clear national guidelines and education and training for MHCPs.

Bergenfeld et al. [44] mentioned that a fiduciary relationship may exist, in which patients trust that their HCP is better informed about their health and consequently transfer their health responsibility to the MHCP [53]. Our qualitative evidence synthesis showed that pregnant women's trust in MHCPs to make decisions about vaccination provides the opportunity to preserve and leverage maternal acceptance for existing and new vaccines. However, the amount of trust that is usually present between pregnant women and MHCPs is influenced by the duration of that relationship and the HCP's knowledge [24]. In addition, how the vaccination discussion is initiated can impact pregnant women's vaccination uptake decision [54]. For instance, the presumptive or motivational communication approach, when supported by evidence, can positively influence pregnant women's decisions regarding the uptake of the influenza vaccine [41,43]. This finding is in agreement with a recent systematic review that studied qualitative and quantitative evidence about factors that influence vaccination decision making among pregnant women; this reported that the odds of influenza vaccine uptake were 10 to 12 times higher among pregnant women who had received a recommendation from their HCPs [55]. Taking informed consent for influenza vaccination from pregnant women might provide a communication opportunity between MHCPs and pregnant women to discuss the vaccination.

The presence of clear national guidlines would enhance MHCPs to discuss and recommend vaccination to pregnant women, as it would promote their responsibility toward the delivery of the vaccination and avoid them of being held liable to the possible adverse events of the vaccination [24,42,47].

Knowledgeable HCPs have shown more confidence discussing and recommending influenza vaccines [24,41,44]. Evidence based and effective training and education is a necessity for MHCPs to be aware of the importance of vaccinations as well as enabling them to discuss any hesitations that they may have, in order to cope with the expectations of the system and the concerns of pregnant women.

**Suggestions for interventions to increase influenza vaccination.**   Although the idea of a vaccination champion was identified by both Frew et al. [43] and Wilson et al. [46], it is not widely applied in clinical practice. Previous studies have shown that vaccination champions can improve vaccination rates [56,57].

The range of MHCPs in the studies in this review were heterogeneous. However, as midwives spend more time with pregnant women than other MHCPs, they are ideally placed to be more engaged with vaccination promotion campaigns and programmes [41,46]. Midwives should receive appropriate vaccination administration training so that pregnant women can receive their vaccines when they are recommended rather than inconveniencing them with extra appointments [41,46].

The WHO initiatives to increase the uptake of influenza vaccination during pregnancy is represented by the global expert group Measuring Behavioural and Social Drivers of Vaccination (BeSD) who has adopted a model for vaccine hesitancy [58]. This model identifies three classifications of hesitancy: a) poor contextual guidance from authority figures, including influential leaders and persons; b) experiences relating to various cases of poor healthcare,

either from the media or within social groups; and c) complications related to vaccines and negative interactions with healthcare providers [58]. The WHO model was designed to help MHCPs tackle vaccination hesitancy. There is overlap between the WHO model and our review findings in terms of how social factors and the media affect pregnant women's decisions to vaccinate. Another overlap is with the need for good communication between MHCP and pregnant women to address pregnant women's concerns about vaccination's safety, which would increase the uptake of maternal influenza vaccine.

## 4.2 Strengths and limitations

Our review used an extensive and comprehensive search of different databases complemented by grey literature search, reference and citation searches; therefore, it is unlikely that published studies would have been missed. Overall, the quality of the included studies was high, although there is a lack of consensus about the assessment of study quality in the synthesis of qualitative research [39].

The use of NVivo software for data coding offers an auditable pathway from the primary data to the results. The data were synthesised from studies representing different contexts and locations, including high-, middle- and low-income countries. Even with this heterogeneity, there was consistency in the findings across contexts, with the analytical themes being described in many of the included studies. In addition, the MHCPs were a heterogeneous group, widening the transferability and providing a range of views and experiences.

Because there are relatively few primary qualitative studies in the field, our review represents the current evidence on MHCPs' views and experiences about influenza vaccination during pregnancy, including their views about sociocultural norms and pregnant women's beliefs about vaccination. Also, there is strong corroboration of evidence across the included studies and wider quantitative studies on the same topic.

However, our review does have some limitations. Non-English language articles were excluded; therefore, the transferability of the findings may be limited [59]. This synthesis represents evidence collected from study locations where the MHCPs' ethnicities were fairly homogenous. Thus, there is the possibility for cultural bias or omissions in our findings, which means that these may not be transferable to different settings and samples. Also, there may be findings that relate to a particular context or setting, in particular settings where access to antenatal care was the underlying cause of lack of access to influenza vaccination, or where professional group's roles and responsibilities varied between settings. The search was limited to studies published after 2012 to reflect the most recent views and experiences of MHCPs following the release of the WHO influenza vaccination guidelines [1].

## 5 Conclusion

Although influenza vaccine could prevent hospitalisation of pregnant women due to influenza infection, provide immunity to the new-born, increasing influenza vaccination uptake in pregnant women remains a challenge. Evidence-informed interventions are needed to improve influenza vaccination rates by addressing MHCPs' hesitations as well as developing or improving local and national guidelines. The suggestions developed in this review may be applicable to the uptake of other vaccines such as Covid-19 vaccine.

## Supporting information

**S1 Table. Preferred Reporting Items for Systematic Reviews and Meta-Analyses (PRISMA) checklist.**
(DOCX)

**S2 Table. Enhancing Transparency of Reporting the Synthesis of Qualitative Research (ENTREQ) checklist.**
(DOCX)

**S3 Table. Example of electronic Boolean search strategy.**
(DOCX)

**S4 Table. Characteristics of included studies.**
(DOCX)

**S5 Table. CASP checklist for included studies.**
(DOCX)

## Author Contributions

**Conceptualization:** Fatemah Alhendyani.

**Data curation:** Fatemah Alhendyani, Laura L. Jones.

**Formal analysis:** Fatemah Alhendyani.

**Funding acquisition:** Fatemah Alhendyani.

**Investigation:** Fatemah Alhendyani.

**Methodology:** Fatemah Alhendyani, Laura L. Jones.

**Project administration:** Fatemah Alhendyani, Laura L. Jones.

**Resources:** Fatemah Alhendyani.

**Software:** Fatemah Alhendyani.

**Supervision:** Kate Jolly, Laura L. Jones.

**Validation:** Fatemah Alhendyani, Laura L. Jones.

**Visualization:** Fatemah Alhendyani, Kate Jolly.

**Writing – original draft:** Fatemah Alhendyani.

**Writing – review & editing:** Fatemah Alhendyani, Kate Jolly, Laura L. Jones.

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
