## [Decision Letter · Decision Letter 0]

4 Jun 2021

PONE-D-21-09606

Views and experiences of maternity healthcare professionals involved in the provision of influenza vaccination during pregnancy: a systematic review and qualitative evidence synthesis

PLOS ONE

Dear Dr. Alhendyani,

Thank you for submitting your manuscript to PLOS ONE. After careful consideration, we feel that it has merit but does not fully meet PLOS ONE’s publication criteria as it currently stands. Therefore, we invite you to submit a revised version of the manuscript that addresses the points raised during the review process.

We look forward to receiving your revised manuscript.

Kind regards,

Emma Sacks

Academic Editor

PLOS ONE

Journal Requirements:

2. Thank you for providing your flowchart as figure S1. Please make this figure 1 in your main manuscript.

3. Please confirm that you have included all items recommended in the PRISMA checklist including the full electronic Boolean search strategy used to identify studies with all search terms and limits for at least one database. Please attach this as supplementary file.

"KJ is supported by the National Institute for Health Research (NIHR) Applied Research

Centre (ARC) West Midlands. The views expressed are those of the author(s) and not

necessarily those of the NIHR or the Department of Health and Social Care."

Additional Editor Comments :

Thank you for your submission of this article on an important and timely topic. However, myself and the reviewer have a number of concerns we are hopeful you can address.

My main concern is that the findings are presented with no weighting, yet most come only from one contributing study, some conflict, and the overall number of included papers is quite small. Consider looking at the guidance for GRADE CERQual methodology and the ways to assess confidence in the review findings (especially if they are not necessarily generalizable to multiple settings). I would suggest adding the contributing studies, along with their quality and location, to Table 3. I don't fully understand why this paper assessed the quality of each contributing study with the CASP but then does not utilize those assessments to eliminate, weight, or contextualize the interpretations of the findings. Instead, the methods include an assessment of data richness, but the assessment of this is not explained.

My other main concern is that the introduction (as well as the findings) need more more differentiation by country/setting or geographic region. The vaccination rates vary widely, and should be discussed. The % of countries which recommend influenza vaccine for pregnant women and if possible, the extent to which national guidance is available, should be included. Further, significant years (introduction of vaccine, change in guidelines, etc.) should be mentioned, and the years of the search strategy should be included.

-In the abstract, there is information in the conclusion not included in the results; new findings should not be presented here

- "decline" would be a better word than "rejection"

-a review of effective interventions would be a different review; if the point here is to ask what MCHPs would recommend, please state that clearly

-Please be careful with the framing as these are MCHPs often reporting on what they THINK is influencing women's uptake of the vaccine. In other research, we have seen that health care workers and patients have different understandings about reasons for patient behaviour.

-What terms are used in the search? Can the search strategy be appended?

-It's a significant limitation to restrict to English-only. How many potential papers were screened out because of language? Please add further explanation in the limitations about the possible biases introduced because of this.

-Why are community health workers included in the results, if they are not included in the inclusion criteria? Please clarify.

-Given how small the sample size is, the findings are far too general. For example, "MHCPs held one on one discussions" is only cited by one study, so cannot be generalized to every practice around the world.

-The headings have reference to the MHCP and women's "characteristics" but I don't see them in the findings (other than the power dynamic between patient and provider, which seems more of a perception and less about the characteristics of the MHCP him/herself)

-"Social and cultural norms" is quite a broad category - as it covers religion, intrahousehold power, and gendered dynamics. Consider separating these out.

-Consider adding identifiers to the quotes - what cadre of health care worker is speaking? From what country was the contributing study?

-It seems that the prohibition on ANC might be one of the most important barriers to vaccination uptake rather than specifics around the vaccine itself. This gets lost in the current framing under social norms, but would certainly require a different set of interventions or messages

-"Media" is pretty broad - what types of media? (tv, internet, radio, social media, etc.)

-gang groups and violence around the clinic seems very specific to one site, do you have evidence that this is more widespread? (see above my comment about CERQual, as this method would downgrade a finding such as this)

-In the discussion, while more confident providers may lead to higher uptake, there was a concerning quote in included in the paper about how patients may not feel empowered to decline something recommended; I would include a caveat here about the importance of informed consent.

-Workload and time constraints feature prominently in the abstract and discussion, but did not come out strongly in the findings as presented (other than deprioritizing this vaccine against others, but there were other confounders)

-Vaccine champions are mentioned but not explained

-I don't understand the phrase "naturally present" with regard to patient-provider relationships; why would this be inherent?

-Table 3 is very unclear: are these the views of MCHPs about women's views?

-Please add line numbers so reviewers can more easily point out suggestions

-there is at least one sentence that is repeated verbatim. Overall, the paper would benefit from a careful proofread, as there are grammatical errors and typos (including in the references and tables)

Reviewers' comments:

Reviewer's Responses to Questions

**Comments to the Author**

1. Is the manuscript technically sound, and do the data support the conclusions?

Reviewer #1: Yes

2. Has the statistical analysis been performed appropriately and rigorously? 

Reviewer #1: N/A

3. Have the authors made all data underlying the findings in their manuscript fully available?

Reviewer #1: Yes

4. Is the manuscript presented in an intelligible fashion and written in standard English?

Reviewer #1: Yes

5. Review Comments to the Author

Reviewer #1: Summary:

This article is systematic review of qualitative studies of MHCPs’ views of and experience with providing influenza vaccines during pregnancy. It then identifies themes from these studies and suggests approaches to enhance vaccination uptake. The authors draw from eight studies, reflecting participation of 277 MCHP’s across low, middle, and high income countries. They identify six analytical themes and 17 descriptive themes and, and from those themes, propose areas for intervention specifically for MHCPs. While socio-cultural norms and the beliefs of pregnant women strongly influence MCHPs’ decision making, barriers of knowledge among MHCPs and structural constraints within the visit, as well as national policy/guidelines and benchmarks are all more intervention-ready targets to increase rates of influenza vaccination in pregnancy. The paper does an adequate job of reviewing available literature and synthesizing themes—the methodology is appropriate for a systematic review.

Comments by section:

Introduction:

- See this recent study Dawood et al, Incidence of influenza during pregnancy and association with pregnancy and perinatal outcomes in three middle-income countries: a multisite prospective longitudinal cohort study - The Lancet Infectious Diseases for additional (though contradictory) data regarding risk of antenatal influenza and birth outcomes

- You can condense your first two sentences, as your readers know pregnant women and their fetuses are at high risk from influenza

- Would make two sentences from Deberra et al study—one about effects of vaccine in pregnant women, one about protection in infants

- Is there a published statistic for uptake of influenza vaccine in pregnancy globally? Looks like about 50% in United States (Low Rates of Vaccination During Pregnancy Leave Moms, Babies Unprotected (cdc.gov))

- Brings up interesting point—there are (at least) two issues at play here: 1) How often are pregnant women being offered the influenza vaccine by their MHCP? 2) How often are women receiving the vaccine? Your paper is primarily concerned with the former, but this is never explicitly delineated.

- When you say “experiences” for MCHPs (2nd sentence, 3 paragraph) do you mean MCHPs’ past experience recommending influenza vaccine? Knowledge, beliefs, and time constraints are fairly self-explanatory concepts, but “experiences” remains unclear to me throughout the paper.

- Your fourth paragraph is strong and makes a good argument for the necessity of the paper.

Methods:

- Well delineated inclusion and exclusion criteria, though would be interesting to comment on difference between MCHPs’ views of Tdap vs influenza vaccines (structural constraints identified as barrier to Tdap vaccination as well) somewhere in discussion. Table 1 very helpful and clear.

- Search and selection strategy well described and sound

Results:

- Table 2 informative

- Quality assessment is useful. Seems like Bergenfeld et al was problematic from recruitment strategy perspective—may need to justify its inclusion (especially since it is heavily referenced in the introduction section)

- Table 3 is clear delineation of themes

- Figure 1. MHCP “characteristics” not clear—do you mean knowledge, attitudes, and beliefs?

- More generally, it seems like there is more to draw out of the “knowledge” and MHCPs. The “Characteristics” and the first part of “Local Healthcare system” sections seem like they would be better combined under “MHCP knowledge and education about influenza vaccine”, especially as this emerges as a target for intervention in your discussion.

- Then the rest of “local healthcare system” is systems factors (checklist, electronic medical record based interventions, place for documentation) which seems like its own theme.

- The record keeping/vaccination history piece of “national policy and practice” is an excellent point, and perhaps should be highlighted in your discussion section as well

Discussion:

- Is there way to explicitly highlight MCHP generated strategies (visiting religious services in Kenya, creating EHR vaccination prompts, dedicated vaccine teams, education hub) to improve influenza vaccination?

- In paragraph 4 of this section, last sentence should become two. This paragraph seems to be the crux of your argument—that MCHPs are uniquely positioned to affect uptake of influenza vaccine.

- For paragraph 6, you don’t really bring this theme out in your thematic analysis in the results section

- For paragraph 7, the vaccination champion idea also is not adequate introduced in the results section

- In paragraph 8, are you suggesting focusing resources on midwives especially? Would make this clear if this is the case.

- In paragraph 9, last sentence is a sentence fragment—would revise

Strengths and Limitations:

- Clear

- Agree that thematic saturation is this study’s greatest strength

Conclusion:

- The last sentence is a run-on and does not make sense.

6. PLOS authors have the option to publish the peer review history of their article (what does this mean?). If published, this will include your full peer review and any attached files.

Reviewer #1: No

---

## [Author Response · Author response to Decision Letter 0]

14 Aug 2021

Editors’ Comment

Comment Response

1. Please ensure that your manuscript meets PLOS ONE's style requirements This has now been amended 

2. Thank you for providing your flowchart as figure S1. Please make this figure 1 in your main manuscript. This now has been added

3. Please confirm that you have included all items recommended in the PRISMA checklist including the full electronic Boolean search strategy used to identify studies with all search terms and limits for at least one database. Please attach this as supplementary file This now has been amended and an example of search strategy has been added as supplementary file

"KJ is supported by the National Institute for Health Research (NIHR) Applied Research Centre (ARC) West Midlands. The views expressed are those of the author(s) and not necessarily those of the NIHR or the Department of Health and Social Care."

We would like to update the funding statement to read:

“The author(s) received no specific funding for this work. KJ is supported by the National Institute for Health Research (NIHR) Applied Research Centre (ARC) West Midlands.”

Reviewer One Comments

Comment Response

1. I would suggest adding the contributing studies, along with their quality and location, to Table 3. The references to the contributing studies have been added to table 3. We do not think that it is necessary to add quality and location to table 3 as this is reported in table 2 and additional details will reduce the clarity of the message.

2. I don't fully understand why this paper assessed the quality of each contributing study with the CASP but then does not utilize those assessments to eliminate, weight, or contextualize the interpretations of the findings. Instead, the methods include an assessment of data richness, but the assessment of this is not explained. Critical appraisal within qualitative systematic reviews is debated within the reviewing methods literature. Few qualitative systematic reviews use a quality appraisal to formally weight studies. We have highlighted in the methods the lack of consensus about the assessment of study quality in the synthesis of qualitative research [42]. Conceptual richness is distinct from methodological reporting quality. Conceptual richness relates to the relevance to the review objectives and how much rich data each paper contributed to the synthesis. A study can be poorly reported but still contribute rich data to the synthesis and vice versa. 

Overall, the reporting quality of the studies was high, this is presented in a supplementary table. We have added additional text to the discussion highlighting the quality of the studies increasing the confidence in the findings. 

“Overall, the quality of the included studies was high, though there is a lack of consensus about the assessment of study quality in the synthesis of qualitative research [42].”

3. The % of countries which recommend influenza vaccine for pregnant women and if possible, the extent to which national guidance is available, should be included. Further, significant years (introduction of vaccine, change in guidelines, etc.) should be mentioned, and the years of the search strategy should be included. We have added the date that WHO recommended flu vaccination in pregnancy (2012). We have also added vaccine coverage rates where these data are available/accessible It’s challenging to find this information. (See page 4)

The dates of the search (2012 to March 20220) were included in the “Search Strategy” section of the Methods. 

4. In the abstract, there is information in the conclusion not included in the results; new findings should not be presented here. This now has been amended 

5. "decline" would be a better word than "rejection" Thank you, we have made that change

6. a review of effective interventions would be a different review; if the point here is to ask what MCHPs would recommend, please state that clearly. We have amended the aim to clarify that it includes identifying interventions that MHCPs consider feasible to implement.

“The aim of the study was to systematically review and thematically synthesise qualitative evidence that explores the views and the experiences of MHCPs involved in the provision of the maternal influenza vaccine worldwide. In addition, we looked for interventions that MHCPs consider feasible to implement to increase vaccination uptake”.

7. Please be careful with the framing as these are MCHPs often reporting on what they THINK is influencing women's uptake of the vaccine. In other research, we have seen that health care workers and patients have different understandings about reasons for patient behaviour. Thank you, we have checked through the manuscript to ensure that we report that the views are from MCHPs, not directly from pregnant women. 

8. What terms are used in the search? Can the search strategy be appended? The key search terms are included within the search strategy section of the methods. We have now added an example search strategy for one database as a supplementary file. 

9. It's a significant limitation to restrict to English-only. How many potential papers were screened out because of language? Please add further explanation in the limitations about the possible biases introduced because of this. The search strategy was restricted to English language only, so we are unable to estimate this. We did undertake a search in Medline to see whether we had missed any non-English studies, but found none. We acknowledge this limitation in the discussion section. 

10. Why are community health workers included in the results, if they are not included in the inclusion criteria? Please clarify. Community health workers were included under the group “health workers and public health personnel working in maternity healthcare settings”. We have now added “e.g., community health workers” to clarify this.

11. Given how small the sample size is, the findings are far too general. For example, "MHCPs held one on one discussions" is only cited by one study, so cannot be generalized to every practice around the world. Communication between MHCP and pregnant women have been mentioned in three studies and we have demonstrated different approaches which are: one-to-one discussion and using passive language [45, 48, 50]. 

12. The headings have reference to the MHCP and women's "characteristics" but I don't see them in the findings (other than the power dynamic between patient and provider, which seems more of a perception and less about the characteristics of the MHCP him/herself) To avoid confusion, we have amended these, so the analytical themes are now: “Pregnant women’s health literacy and beliefs” and “MHCPs’ knowledge and attitudes”.

13. “Social and cultural norms" is quite a broad category - as it covers religion, intrahousehold power, and gendered dynamics. Consider separating these out. We have now amended these, so the analytical theme is now: “social, cultural and environmental influences” and descriptive themes are now: “religion, family influences, media influence and access issues”

14. Consider adding identifiers to the quotes - what cadre of health care worker is speaking? From what country was the contributing study? We have added the country. We have also added whether it was a primary quote (italics) or author interpretation (normal font). It is not possible to add the cadre of staff as not all primary quotes were identified in paper and some quotes are author interpretations. 

15. It seems that the prohibition on ANC might be one of the most important barriers to vaccination uptake rather than specifics around the vaccine itself. This gets lost in the current framing under social norms, but would certainly require a different set of interventions or messages Please see our response to point 13. We have now amended these, so the analytical theme is now: “social, cultural and environmental influences” and descriptive themes are now: “religion, family influences, media influence and access issues”. 

16. "Media" is pretty broad - what types of media? (tv, internet, radio, social media, etc.) We are unable to add additional detail and this is not mentioned/ specified in the study

17. gang groups and violence around the clinic seems very specific to one site, do you have evidence that this is more widespread? (see above my comment about CERQual, as this method would downgrade a finding such as this) Please see our response to point 2. We agree that this is an issue limited to one study, but it fits within the wider issues relating the access issues.

18. In the discussion, while more confident providers may lead to higher uptake, there was a concerning quote in included in the paper about how patients may not feel empowered to decline something recommended; I would include a caveat here about the importance of informed consent. We agree with the reviewer, now we have added this:

 “It is, however, important that MHCPs support informed consent for influenza vaccination”

19. Workload and time constraints feature prominently in the abstract and discussion, but did not come out strongly in the findings as presented (other than deprioritizing this vaccine against others, but there were other confounders) We have added additional results and quotes to support this:

“Workload and time limit were factors reported by MHCPs limiting influenza vaccination discussion with pregnant women [22, 50]:

 “Maybe when a health care provider is in a hurry or is being overworked, you may find a long queue at the ANC waiting for vaccination. The nurse there may not have time to discuss much with every client about the vaccines. Sometimes they issue orders for the mothers to queue and get vaccinated. These are situations which may happen when there are several mothers at the clinic. This can cripple vaccine uptake since there is no time for explanations” [Kenya, 22]

“The doctors are very busy and so for example in the private clinic we run at capacity so we’re turning women away, so we basically have to apply um err almost triage principles to how we run our consultations but we do have a um midwife there with us who is our personal sort of assistant if you like. So things like breastfeeding and err analgesia in labor and vaccinations although I don’t know if they mention vaccinations I’ll be honest. We tend to delegate to them. The longer we make the consultations basically the less patients we can see.” [Australia, 50]

20. Vaccine champions are mentioned but not explained This has now been explained:

The need for a vaccination champion who is managing and creating enthusiasm for vaccination campaigns as well as encourages better communication about vaccination between HCPs,…

21. I don't understand the phrase "naturally present" with regard to patient-provider relationships; why would this be inherent? We have amended this term to make the meaning clearer. We now say: “usually present”

22. Table 3 is very unclear: are these the views of MCHPs about women's views? Table 3 presents the key themes which summarise MHCPs’ views about influenza vaccination in pregnancy and what they think are the barriers to women being vaccinated. We have clarified this in the overall aim and re-worded the table caption which now reads:

“Core analytical and descriptive themes for MHCPs’ experiences and views relating to the provision of influenza vaccination for pregnant women”

23. Please add line numbers so reviewers can more easily point out suggestions Line numbers have been added

Reviewer Two Comments

Comment Response

Introduction

24. See this recent study Dawood et al, Incidence of influenza during pregnancy and association with pregnancy and perinatal outcomes in three middle-income countries: a multisite prospective longitudinal cohort study - The Lancet Infectious Diseases for additional (though contradictory) data regarding risk of antenatal influenza and birth outcomes. Thank you – we have added detail about this study.

25. You can condense your first two sentences, as your readers know pregnant women and their fetuses are at high risk from influenza Thank you – we have made this change

26. Would make two sentences from Deberra et al study—one about effects of vaccine in pregnant women, one about protection in infants Thank you – we have made this change

27. Is there a published statistic for uptake of influenza vaccine in pregnancy globally? Looks like about 50% in United States (Low Rates of Vaccination During Pregnancy Leave Moms, Babies Unprotected (cdc.gov))

We have added this statistic for several countries but there does not appear to be a global statistic readily accessible

“The two recommended vaccines during pregnancy are influenza vaccine and Tdap (Tetanus, diphtheria, and pertussis) vaccine. However, in 2019–20 in the United States, 61.2% of pregnant women received influenza vaccine, 56.6% received Tdap, and 40.3% received both vaccines [64]. A 2020 expert commentary review reported an 8.7% median influenza vaccine coverage rate in pregnant women in the European region [63], whereas in the US during the 2017–2018 influenza season, they reported a 49.1% influenza vaccine uptake before or during pregnancy. High rates have been achieved, with 95.7% uptake in Brazil in 2010 as a result of a government vaccination campaign [63].”

28. Brings up interesting point—there are (at least) two issues at play here: 1) How often are pregnant women being offered the influenza vaccine by their MHCP? 2) How often are women receiving the vaccine? Your paper is primarily concerned with the former, but this is never explicitly delineated. We agree with the reviewer that there are potentially separate issues, although uptake/receipt of the vaccine is dependent on it being offered. We have drawn out this distinction in the introduction:

“Questions such as how often pregnant women are being offered the influenza vaccine or how often pregnant women are receiving the influenza vaccine need to be addressed.”

29. When you say “experiences” for MCHPs (2nd sentence, 3 paragraph) do you mean MCHPs’ past experience recommending influenza vaccine? Knowledge, beliefs, and time constraints are fairly self-explanatory concepts, but “experiences” remains unclear to me throughout the paper. We use the term ‘experiences’ within the paper to include MHCPs’ experiences with pregnant women in relation to discussing and delivering maternal influenza vaccine.

30. Your fourth paragraph is strong and makes a good argument for the necessity of the paper. Thank you

Methods

31. Well delineated inclusion and exclusion criteria, though would be interesting to comment on difference between MCHPs’ views of Tdap vs influenza vaccines (structural constraints identified as barrier to Tdap vaccination as well) somewhere in discussion. The latest CDC statistics about Influenza and Tdap coverage rate is now added to the introduction. Whilst we agree that the contrast between the barriers for delivering Tdap and influenza in pregnancy is interesting, we feel that it would distract from the clarity of the narrative in relation to influenza vaccination. 

Result

32. Table 2 informative Thank you

33. Quality assessment is useful. Seems like Bergenfeld et al was problematic from recruitment strategy perspective—may need to justify its inclusion (especially since it is heavily referenced in the introduction section) Please see response to point 2 above. We didn’t specify any eligibility criteria in relation to quality of studies, and Bergenfield otherwise had high quality reporting, so we believe it was correct to include the study. 

34. Table 3 is clear delineation of themes Thank you

35. Figure 1. MHCP “characteristics” not clear—do you mean knowledge, attitudes, and beliefs? To avoid confusion, we have amended these, so the analytical theme is now “MHCPs’ knowledge and attitudes”. See response to point 13 above. 

36. More generally, it seems like there is more to draw out of the “knowledge” and MHCPs. The “Characteristics” and the first part of “Local Healthcare system” sections seem like they would be better combined under “MHCP knowledge and education about influenza vaccine”, especially as this emerges as a target for intervention in your discussion. We carefully considered the themes. However, as with all qualitative research and thematic-syntheses, different research teams might draw out different analytical and sub-themes. We have addressed this point with the one below. 

37. Then the rest of “local healthcare system” is systems factors (checklist, electronic medical record-based interventions, place for documentation) which seems like its own theme. 

Thank you for the thoughtful suggestion. We agree that training and education could be considered to fit within ‘local health care system’ or ‘MCHP’s knowledge and attitudes’. We would like to keep the training and education within the local healthcare system as the delivery of this is not something that should be the responsibility of an individual MHCP, but should be considered for the whole maternity health care system as a part of organised updates and ongoing training and development. 

38. The record keeping/vaccination history piece of “national policy and practice” is an excellent point, and perhaps should be highlighted in your discussion section as well Thank you – we have now included this point within the discussion:

Discussion

39. Is there way to explicitly highlight MCHP generated strategies (visiting religious services in Kenya, creating EHR vaccination prompts, dedicated vaccine teams, education hub) to improve influenza vaccination? This has now been added:

“Moreover, MHCPs discussed strategies such visiting religious services in Kenya,..”

40. In paragraph 4 of this section, last sentence should become two. This paragraph seems to be the crux of your argument—that MCHPs are uniquely positioned to affect uptake of influenza vaccine. We have split the sentence with a semi-colon to clarify the meaning. We agree that this is an important finding.

41. For paragraph 6, you don’t really bring this theme out in your thematic analysis in the results section. Please see response to point 19 above.

42. For paragraph 7, the vaccination champion idea also is not adequate introduced in the results section Please see response to point 20 above.

43. In paragraph 8, are you suggesting focusing resources on midwives especially? Would make this clear if this is the case. The paragraph does suggest that midwives would be well placed for this role. 

44. In paragraph 9, last sentence is a sentence fragment—would revised This sentence has been rephrased to improve its clarity.

Conclusion

45. The last sentence is a run-on and does not make sense. The sentence has been re-written:

Evidence-informed interventions would be needed to improve influenza vaccination rates by addressing both pregnant women’s and MHCPs hesitations as well as developing or improving local and national guidelines.

---

## [Decision Letter · Decision Letter 1]

25 Oct 2021

PONE-D-21-09606R1Views and experiences of maternity healthcare professionals involved in the provision of influenza vaccination during pregnancy: a systematic review and qualitative evidence synthesisPLOS ONE

Dear Dr. Alhendyani,

Thank you for submitting your manuscript to PLOS ONE. After careful consideration, we feel that it has merit but does not fully meet PLOS ONE’s publication criteria as it currently stands. Therefore, we invite you to submit a revised version of the manuscript that addresses the points raised during the review process.

We look forward to receiving your revised manuscript.

Kind regards,

Emma Sacks

Academic Editor

PLOS ONE

Journal Requirements:

Additional Editor Comments (if provided):

Thank you for your revisions to this manuscript. Reviewer #1 still has some additional minor comments we are hopeful you can address.

I also still have some concerns that less common findings (e.g. gang violence) is presented without qualification of how common it might be compared to other findings. While there is no expectation of quantification in qualitative papers, it would still be helpful to emphasise which findings were most common across papers, and which were identified in fewer, specific contexts.

Reviewers' comments:

Reviewer's Responses to Questions

**Comments to the Author**

1. If the authors have adequately addressed your comments raised in a previous round of review and you feel that this manuscript is now acceptable for publication, you may indicate that here to bypass the “Comments to the Author” section, enter your conflict of interest statement in the “Confidential to Editor” section, and submit your "Accept" recommendation.

Reviewer #1: All comments have been addressed

2. Is the manuscript technically sound, and do the data support the conclusions?

Reviewer #1: Yes

3. Has the statistical analysis been performed appropriately and rigorously? 

Reviewer #1: N/A

4. Have the authors made all data underlying the findings in their manuscript fully available?

Reviewer #1: Yes

5. Is the manuscript presented in an intelligible fashion and written in standard English?

Reviewer #1: Yes

6. Review Comments to the Author

Reviewer #1: Global comments:

--Most of section 4.5.1 relates to barriers to ANC in general, not vaccines in particular

--Would benefit from more consistent language throughout. The paper focuses on the perspectives of MHCP’s on the influenza vaccine (rather than knowledge, attitude, and beliefs). Among themes emerging from the qualitative studies reviewed are “social, cultural, and environmental influences,” “MHCP’s knowledge and belief” about the vaccines. This extends to the title—might try “Perceptions of Maternal Health Care Providers on …”

--The addition of country and study to the included quotes is very helpful.

--Still don’t see much critical evaluation of studies included in your discussion section—for example, issues around the heterogeneity of providers included makes the discussion of who is responsible for advocating for influenza vaccine confusing (MD/DO’s believing it is a nurse role)

--Would benefit from more classification of findings in discussion: on barriers (workload, time constraints, MHCP’s perceptions of women’s concerns about flu vaccine, lack of knowledge on the part of MHCP re: flu vaccine), facilitators (good rapport, national guidelines, education/training) and suggestions for further intervention (vaccine champions, EMR prompts, systems level informatics regarding who is due for flu shot, free vaccines)

--This is a very timely study given the applicability to COVID vaccines in pregnancy—would probably be more explicit about this when it comes to significance—understanding the perspective of MHCP’s on influenza vaccine may shape strategy for educating them about COVID vaccine in pregnancy.

Line by line:

26-28 Still unclear: “experience, knowledge, beliefs” of the MHCP’s themselves? Still vague—would reword for clarity, especially in abstract. This is broken down clearly in your thematic analysis so why not say “knowledge and attitude” and “social, cultural, and environmental influences.”

41-42 Same—are you saying that their commitment to provision of flu vaccine was challenged by workload and time constraints? By social, cultural, and environmental influences? By women’s perceived concerns?

Perhaps cast these as facilitators: trusting relationships, good communication, knowledge about the vaccine leading to confidence in recommending vaccine, EMR prompts, and presence of national guidelines. And then the barriers: workload, time constraints, MHCP’s perception of pregnant women’s concerns, and social/cultural/environmental influences.

71 Don’t need “however”

83-86 should read: Evidence showed the vaccine offered immunity

90 I think you mean: In 2019-2020 in the US, 61.2% of pregnant women received influenza vaccine, 56.6% received Tdap, and 40.3 received both.

89-96 These statistics seem contradictory—may need a sentence up front suggesting “….heterogeneity globally—from less than 10% coverage in Europe, 50-60% in US, and >95% in Brazil.”

98-101 likely a typo here affecting clarity: I think you are saying “MHCPs knowledge of and beliefs about the influenza vaccine, and their experiences supporting pregnant women, as well as their workload and time constraints, affect their practices in advocating for influenza vaccine during pregnancy. “

109-110 These are fundamentally different questions, and your study seems to focus on the factors affecting the first of these. It’s an interesting and important point—maybe in focusing on pregnant women’s perspectives, we’re actually missing one of the main problems, which is that they are not being offered (or encouraged to take) the influenza vaccine. It follows then that is essential to understand the perspective of those offering/recommending the vaccine.

178 need a period at the end of this sentence.

198 will need to comment on the heterogeneity of this group in your discussion section

235 4.5.1 None of these particular to vaccine uptake, they all seem more generally related to provision of ante-natal care. (257-260 reference is the exception here). This is obviously not prohibitive in including this section, but would just acknowledge that the barriers here are largely barriers to ANC itself.

374 this is where the heterogeneity of your population is problematic—some of your MHCP are nurses

386 need semi colon between studies; it

386 they is unclear. Would say “vaccine champions” may increase

395: setting a national policy and guidelines, centralizing vaccine information, and providing influence vaccine for free would enhance…

430-433 these need separate sentences—they are all very different ideas.

433 take out “likely”

446 Would re-word for clarity

452 should read “for existing and new vaccines”

467-470 run on sentence

471 don’t need “however”

481 maybe discussion of heterogeneity of providers included in this study goes here, and then highlight role of midwives

484 take out comma after pregnancy

491 YES! Lots of overlap. This helps add validity to your findings (or WHO’s model…)

Figure 2 I like this model

7. PLOS authors have the option to publish the peer review history of their article (what does this mean?). If published, this will include your full peer review and any attached files.

Reviewer #1: No

---

## [Author Response · Author response to Decision Letter 1]

8 Dec 2021

Thank you for the opportunity to make changes to strengthen this manuscript. We thank the reviewer for their helpful and detailed comments which we have addressed below. Line numbers refer to the clean untracked version.

Reviewer’s global comments

 Reviewer comment Author response

1 Most of section 4.5.1 relates to barriers to ANC in general, not vaccines in particular The included studies were only concerned about vaccine during pregnancy including influenza, so the statement reflects facts about influenza vaccine and other pregnancy vaccines as well. However, we have amended a sentence at the start of section 4.5.1 (line 235-238) to make this clearer that lack of access to care influences the ability to access the vaccine:

“Social and cultural norms, including religion, family involvement, media influence and access issues, were perceived by MHCPs to influence a pregnant women’s decision about influenza vaccination in pregnancy or access to antenatal care and thus the opportunity to be offered influenza vaccination.”

2 Would benefit from more consistent language throughout. The paper focuses on the perspectives of MHCP’s on the influenza vaccine (rather than knowledge, attitude, and beliefs). Among themes emerging from the qualitative studies reviewed are “social, cultural, and environmental influences,” “MHCP’s knowledge and belief” about the vaccines. This extends to the title—might try “Perceptions of Maternal Health Care Providers on …” We have also tried to be consistent with language using views and experiences, rather than knowledge, attitudes and beliefs.

This is now consistent with the title: “Views and experiences of maternal healthcare providers on influenza vaccine during pregnancy: a systematic review and qualitative evidence synthesis”

3 The addition of country and study to the included quotes is very helpful. Thank you

4 Still don’t see much critical evaluation of studies included in your discussion section—for example, issues around the heterogeneity of providers included makes the discussion of who is responsible for advocating for influenza vaccine confusing (MD/DO’s believing it is a nurse role) The heterogeneity of MHCPs is now addressed in several places within the discussion (see responses to points 16, 18 and 28). We have added a sentence near the start of the discussion that raises the issues of boundary issues between different MHCPs (line 426-428):

“The range of different MHCPs who could be involved in promoting and delivering influenza vaccination can lead to boundary issues as to whose responsibility it is.”

5 Would benefit from more classification of findings in discussion: on barriers (workload, time constraints, MHCP’s perceptions of women’s concerns about flu vaccine, lack of knowledge on the part of MHCP re: flu vaccine), facilitators (good rapport, national guidelines, education/training) and suggestions for further intervention (vaccine champions, EMR prompts, systems level informatics regarding who is due for flu shot, free vaccines) The discussion has been re-ordered so that a discussion of the barriers, facilitators and suggestions for interventions to increase vaccine uptake are explicitly covered within individual headings in the context of the wider literature. 

6 This is a very timely study given the applicability to COVID vaccines in pregnancy—would probably be more explicit about this when it comes to significance—understanding the perspective of MHCP’s on influenza vaccine may shape strategy for educating them about COVID vaccine in pregnancy. In the discussion (line 435-437) we have added:

“Our findings may have insights for the provision of vaccination against COVID-19 during pregnancy such as knowledge and beliefs perceived by the MHCPs toward the delivery of COVID-19 vaccination.”

Reviewer’s line by line comments

 Reviewer comment Authors response

7 26-28 Still unclear: “experience, knowledge, beliefs” of the MHCP’s themselves? Still vague—would reword for clarity, especially in abstract. This is broken down clearly in your thematic analysis so why not say “knowledge and attitude” and “social, cultural, and environmental influences.” This now has been amended as (line 27-29):

However, factors such as MHCPs’ views and knowledge about the vaccine, and time constraints due to workload may influence MHCPs’ practices and opinions about women receiving the influenza vaccine during pregnancy.

8 41-42 Same—are you saying that their commitment to provision of flu vaccine was challenged by workload and time constraints? By social, cultural, and environmental influences? By women’s perceived concerns?

Perhaps cast these as facilitators: trusting relationships, good communication, knowledge about the vaccine leading to confidence in recommending vaccine, EMR prompts, and presence of national guidelines. And then the barriers: workload, time constraints, MHCP’s perception of pregnant women’s concerns, and social/cultural/environmental influences. This now has been amended as (line 42-48):

“…facilitated by trusting relationships, good communication, knowledge about the vaccine leading to confidence in recommending vaccine, EMR prompts, and presence of national guidelines. However, workload, time constraints, MHCP’s perception of pregnant women’s concerns, and social/cultural/environmental influences could prevent the likelihood of delivery of influenza vaccine.”

9 71 Don’t need “however” This now has been amended

10 83-86 should read: Evidence showed the vaccine offered immunity This now has been amended

11 90 I think you mean: In 2019-2020 in the US, 61.2% of pregnant women received influenza vaccine, 56.6% received Tdap, and 40.3 received both This now has been amended

12 89-96 These statistics seem contradictory—may need a sentence up front suggesting “….heterogeneity globally—from less than 10% coverage in Europe, 50-60% in US, and >95% in Brazil.” This now has been amended as: (line 91-94)

“In 2019–20 the United States (US) reported that 61.2% of pregnant women received influenza vaccine, 56.6% received Tdap, and 40.3% received both vaccines [19]. A 2020 expert commentary review reported heterogeneity of coverage globally, from less than 10% coverage in Europe, to 50-60% in US, and >95% in Brazil [20].”

13 98-101 likely a typo here affecting clarity: I think you are saying “MHCPs knowledge of and beliefs about the influenza vaccine, and their experiences supporting pregnant women, as well as their workload and time constraints, affect their practices in advocating for influenza vaccine during pregnancy. “ This now has been amended

14 109-110 These are fundamentally different questions, and your study seems to focus on the factors affecting the first of these. It’s an interesting and important point—maybe in focusing on pregnant women’s perspectives, we’re actually missing one of the main problems, which is that they are not being offered (or encouraged to take) the influenza vaccine. It follows then that is essential to understand the perspective of those offering/recommending the vaccine. We have tried to clarify the focus of the review in relation to the questions (line 108-110):

“Questions such as how often pregnant women are being offered the influenza vaccine or how often pregnant women are receiving the influenza vaccine need to be addressed. This review largely focusses on the offer of the vaccine. The aim of the study was to systematically review and….”

15 178 need a period at the end of this sentence. 

Added, thank you

16 198 will need to comment on the heterogeneity of this group in your discussion section We have added a sentence to the discussion section (line 517-518):

“In addition, the MHCPs were a heterogeneous group, widening the transferability and providing a range of views and experiences.”

17 235 4.5.1 None of these particular to vaccine uptake, they all seem more generally related to provision of ante-natal care. (257-260 reference is the exception here). This is obviously not prohibitive in including this section, but would just acknowledge that the barriers here are largely barriers to ANC itself We have amended a sentence at the start of section 4.5.1 to make this clearer that lack of access to care influences the ability to access the vaccine (line 235-238):

“Social and cultural norms, including religion, family involvement, media influence and access issues, were perceived by MHCPs to influence a pregnant women’s decision about influenza vaccination in pregnancy or access to antenatal care and thus the opportunity to be offered influenza vaccination.”

We have also highlighted this relationship within the discussion section (limitations – lines 522-). 

“Also, there may be findings that relate to a particular context or setting, in particular settings where access to antenatal care was the underlying cause of lack of access to influenza vaccination.”

18 374 this is where the heterogeneity of your population is problematic—some of your MHCP are nurses Please see earlier response about heterogeneity. We have also added it as a limitation within the discussion section (lines 528-531):

“Also, there may be findings that relate to a particular context or setting, in particular settings where access to antenatal care was the underlying cause of lack of access to influenza vaccination, or where professional group’s roles and responsibilities varied between settings”

19 386 need semi colon between studies; it This now has been amended

20 386 they is unclear. Would say “vaccine champions” may increase This now has been amended

21 395: setting a national policy and guidelines, centralizing vaccine information, and providing influence vaccine for free would enhance… This now has been amended

22 430-433 these need separate sentences—they are all very different ideas. We have split this into two sentences to address knowledge and uptake and delivery as separate issues (lines 430-435):

“Moreover, MHCPs discussed strategies to increase knowledge and vaccine uptake, such as visiting religious services in Kenya, and creating educational hubs for pregnant women. Suggestions to increase delivery of the vaccine included creating electronic vaccination prompts and a dedicated vaccine team.”

23 433 take out “likely” This now has been amended

24 446 Would re-word for clarity This now has been amended to (line 477-479): 

“Taking informed consent for influenza vaccination from pregnant women might provide a communication opportunity between MHCPs and pregnant women to discuss the vaccination.”

25 452 should read “for existing and new vaccines” This now has been amended

26 467-470 run on sentence This is now has been amended as (line 481-484): 

“Evidence based and effective training and education is a necessity for MHCPs to be aware of the importance of vaccinations as well as enabling them to discuss any hesitations that they may have, in order to cope with the expectations of the system and the concerns of pregnant women.”

27 471 don’t need “however” done

28 481 maybe discussion of heterogeneity of providers included in this study goes here, and then highlight role of midwives Thank you for the suggestion. It now reads (line 489-491):

“The range of MHCPs in the studies in this review were heterogeneous. However, as midwives spend more time with pregnant women than other MHCPs, they are ideally placed to be more engaged with vaccination promotion campaigns and programmes.”

29 484 take out comma after pregnancy done

30 491 YES! Lots of overlap. This helps add validity to your findings (or WHO’s model…)

Figure 2 I like this model Thank you

---

## [Decision Letter · Decision Letter 2]

24 Dec 2021

PONE-D-21-09606R2Views and experiences of maternal healthcare providers on influenza vaccine during pregnancy: a systematic review and qualitative evidence synthesisPLOS ONE

Dear Dr. Alhendyani,

Thank you for submitting your manuscript to PLOS ONE. After careful consideration, we feel that it has merit but does not fully meet PLOS ONE’s publication criteria as it currently stands. Therefore, we invite you to submit a revised version of the manuscript that addresses the points raised during the review process.

We look forward to receiving your revised manuscript.

Kind regards,

Emma Sacks

Academic Editor

PLOS ONE

Journal Requirements:

Additional Editor Comments (if provided):

Thanks to the authors for their revisions. Both myself and the reviewer have some additional minor comments.

-My biggest concern is that the "views and perspectives" column in Table 3 is not clear. The phrases are too short and the table does not standalone.

-I am still concerned that describing coercion or massive power imbalances are described as good rapport (line 306)

-The very end indicates that there is a start date to the search strategy, but this is not included in the methods

-There may not be consensus about how to assess quality of qualitative papers, but there are methods, including eliminating ones not meeting a certain threshold

-I'd still like to see more differentiation between findings from many papers verses one or few papers, and perhaps a bit more clarity between HCP's words themselves and what is ascribed to them

-One strength of this paper is the global nature; consider adding "global" to the title

-EMR should be spelled out in the abstract

-line 113 - "looked for" is not very specific (perhaps identify and analyse?)

-table 1: what about health care providers who are also pregnant?

-line 190 should be "represented"

-GP should be spelled out at first use

-line 262 should be "including"

-line 289: literacy is not the same as materials (literacy is a skill)

-line 357: should be "improved/strengthened" recordkeeping (record keeping should already be happening)

-line 384: should be MHCP

Reviewers' comments:

Reviewer's Responses to Questions

**Comments to the Author**

1. If the authors have adequately addressed your comments raised in a previous round of review and you feel that this manuscript is now acceptable for publication, you may indicate that here to bypass the “Comments to the Author” section, enter your conflict of interest statement in the “Confidential to Editor” section, and submit your "Accept" recommendation.

Reviewer #1: All comments have been addressed

2. Is the manuscript technically sound, and do the data support the conclusions?

Reviewer #1: Yes

3. Has the statistical analysis been performed appropriately and rigorously? 

Reviewer #1: N/A

4. Have the authors made all data underlying the findings in their manuscript fully available?

Reviewer #1: Yes

5. Is the manuscript presented in an intelligible fashion and written in standard English?

Reviewer #1: Yes

6. Review Comments to the Author

Reviewer #1: The overall organization is much clearer, especially discussion separated into barriers, facilitators, and suggestions to improve vaccination rate.

Some specific line by line comments:

Title—Would consider “regarding” or “of” rather than “on” influenza vaccination

308 to 311—these are different aspects of the conversation—the 1:1 nature and allowing women to ask questions, and then the use of passive language. Would separate into two sentences and discuss implications of each. Passive language sounds like it belongs in lines 314-318 re emphasizing that is the woman’s choice

423-424 think more “attitudes” than “views”

Think casting as barriers and facilitators is makes for much clearer delineation of the findings from the studies

461 “trust and good rapport between MHCP and patient (or pregnant woman), clear national guidelines, and education and training for MHCPs

463-479 Seems like you need to discuss national guidelines in this section, since you included it among facilitators

485 would bold or italicize to mark this as a separate section

538 Would emphasize the importance of the influenza vaccine again here, protecting both mother and fetus.

Could also suggest there may be implications for uptake of other, newer vaccines

7. PLOS authors have the option to publish the peer review history of their article (what does this mean?). If published, this will include your full peer review and any attached files.

Reviewer #1: No

---

## [Author Response · Author response to Decision Letter 2]

11 Jan 2022

Editor’s comments

Editor comment Authors response

Please review your reference list to ensure that it is complete and correct. If you have cited papers that have been retracted, please include the rationale for doing so in the manuscript text, or remove these references and replace them with relevant current references. Any changes to the reference list should be mentioned in the rebuttal letter that accompanies your revised manuscript. If you need to cite a retracted article, indicate the article’s retracted status in the References list and also include a citation and full reference for the retraction notice. Please be informed that no articles were retracted nor were there any changes to the reference list of the manuscript.

My biggest concern is that the "views and perspectives" column in Table 3 is not clear. The phrases are too short and the table does not standalone. Column renamed to “codes” to use the same jargon as Thomas and Harden. Phrases are short here because these were the codes used in the thematic synthesis process as per Thomas and Harden. The purpose of the table is to provide a bird's eye view of the synthesis and themes, each of these codes and themes are described in greater detail in the findings below (this is explained in the newly added line 218)

I am still concerned that describing coercion or massive power imbalances are described as good rapport (line 306) Thank you for this possible interpretation of the selected quote, Bergenfeld et al’s interpretation of the quote was that the social position and knowledge of MHCPs was part of why mother trusted MHCPs and deferred vaccination decisions to the advice to the MHCP. This interpretation by the original author was made clearer by editing and moving the sentence “Bergenfeld et al. explained that….” (Line 301-303)

The very end indicates that there is a start date to the search strategy, but this is not included in the methods This was clarified in the edits to line124-125

-There may not be consensus about how to assess quality of qualitative papers, but there are methods, including eliminating ones not meeting a certain threshold Quality assessment was described. CASP checklists were used and appended. There were no majorly flawed papers as described in the manuscript. The decision of the authors was to not exclude any qualitative papers as all papers contributed to the understanding of the perspectives. Please refer to line 158-159.

-I'd still like to see more differentiation between findings from many papers verses one or few papers, and perhaps a bit more clarity between HCP's words themselves and what is ascribed to them Table 3 shows a clear picture of how many papers contributed to each finding. In the explanation of each paper there is clear indication of the number of papers that contributed to each finding as well from the citation and writing. A line explaining the table contents was added (Line 218).

All quotes included are directly from HCP participants in the respective papers. This was clarified in the following addition in line 218-219.

-One strength of this paper is the global nature; consider adding "global" to the title This now has been amended to “Views and experiences of maternal healthcare providers regarding influenza vaccine during pregnancy globally: a systematic review and qualitative evidence synthesis”

-EMR should be spelled out in the abstract This now has been amended

-line 113 - "looked for" is not very specific (perhaps identify and analyse?) This now has been amended

-table 1: what about health care providers who are also pregnant? This now has been amended 

-line 190 should be "represented" This now has been amended

-GP should be spelled out at first use This now has been amended

-line 262 should be "including" This now has been amended

-line 289: literacy is not the same as materials (literacy is a skill) This now has been amended to “materials”

-line 357: should be "improved/strengthened" recordkeeping (record keeping should already be happening) This now has been amended

-line 384: should be MHCP This now has been amended

Reviewer’s comments

Reviewer comments Authors responses

Title—Would consider “regarding” or “of” rather than “on” influenza vaccination This now has been amended as” Views and experiences of maternal healthcare providers regarding influenza vaccine during pregnancy globally: a systematic review and qualitative evidence synthesis”

308 to 311—these are different aspects of the conversation—the 1:1 nature and allowing women to ask questions, and then the use of passive language. Would separate into two sentences and discuss implications of each. Passive language sounds like it belongs in lines 314-318 re-emphasizing that is the woman’s choice Thank you for this comment,

1-1 discussion was further explained with the inclusion of a quote to depict this. Please refer to line 306-313.

New paragraph was created to explain the use of passive language. A line was added to explain how the use of passive language proposes an option/choice for mothers (line 314-318)

423-424 think more “attitudes” than “views” This now has been amended, thank you

Think casting as barriers and facilitators is makes for much clearer delineation of the findings from the studies Headings were added to delineate barriers and facilitators more clearly 

461 “trust and good rapport between MHCP and patient (or pregnant woman), clear national guidelines, and education and training for MHCPs This now has been amended, thank you

463-479 Seems like you need to discuss national guidelines in this section, since you included it among facilitators National guideline was added to this section in line 490-492

485 would bold or italicize to mark this as a separate section This now has been amended, thank you

538 Would emphasize the importance of the influenza vaccine again here, protecting both mother and fetus. This now has been amended as” Although influenza vaccine could prevent hospitalisation of pregnant women due to influenza infection, provide immunity to the new-born, increasing influenza vaccination uptake in pregnant women remains a challenge.”

Could also suggest there may be implications for uptake of other, newer vaccines This now has been amended as” The suggestions developed in this review may be applicable to the uptake of other vaccines such as Covid-19 vaccine.”

---

## [Editor Report · Decision Letter 3]

18 Jan 2022

Views and experiences of maternal healthcare providers regarding influenza vaccine during pregnancy globally: a systematic review and qualitative evidence synthesis

PONE-D-21-09606R3

Dear Dr. Alhendyani,

We’re pleased to inform you that your manuscript has been judged scientifically suitable for publication and will be formally accepted for publication once it meets all outstanding technical requirements.

Kind regards,

Emma Sacks

Academic Editor

PLOS ONE

Additional Editor Comments (optional):

Thank you for your multiple iterations to improve this paper. We thank you for this important contribution to the literature.

---

## [Editor Report · Acceptance letter]

2 Feb 2022

PONE-D-21-09606R3 

Views and experiences of maternal healthcare providers regarding influenza vaccine during pregnancy globally: a systematic review and qualitative evidence synthesis 

Dear Dr. Alhendyani:

I'm pleased to inform you that your manuscript has been deemed suitable for publication in PLOS ONE. Congratulations! Your manuscript is now with our production department. 

Kind regards, 

on behalf of

Dr. Emma Sacks 

Academic Editor

PLOS ONE